# Protamine Characterization by Top-Down Proteomics: Boosting Proteoform Identification with DBSCAN

**DOI:** 10.3390/proteomes9020021

**Published:** 2021-04-30

**Authors:** Gianluca Arauz-Garofalo, Meritxell Jodar, Mar Vilanova, Alberto de la Iglesia Rodriguez, Judit Castillo, Ada Soler-Ventura, Rafael Oliva, Marta Vilaseca, Marina Gay

**Affiliations:** 1Institute for Research in Biomedicine (IRB Barcelona), The Barcelona Institute of Science and Technology, Baldiri Reixac, 10, 08028 Barcelona, Spain; gianluca.arauz@irbbarcelona.org (G.A.-G.); mar.vilanova@irbbarcelona.org (M.V.); 2Molecular Biology of Reproduction and Development Research Group, Institut d’Investigacions Biomèdiques August Pi i Sunyer (IDIBAPS), Fundació Clínic per a la Recerca Biomèdica (FCRB), Faculty of Medicine and Health Sciences, University of Barcelona, 08036 Barcelona, Spain; meritxell.jodar@ub.edu (M.J.); adelaiglesia@ub.edu (A.d.l.I.R.); juditcastillo@ub.edu (J.C.); ada.soler.ventura@gmail.com (A.S.-V.); 3Biochemistry and Molecular Genetics Service, Hospital Clínic de Barcelona, 08036 Barcelona, Spain

**Keywords:** male infertility, sperm, protamine, LC-MS/MS, top-down proteomics, proteoform, post-translational modifications, Bioinformatics, DBSCAN

## Abstract

Protamines replace histones as the main nuclear protein in the sperm cells of many species and play a crucial role in compacting the paternal genome. Human spermatozoa contain protamine 1 (P1) and the family of protamine 2 (P2) proteins. Alterations in protamine PTMs or the P1/P2 ratio may be associated with male infertility. Top-down proteomics enables large-scale analysis of intact proteoforms derived from alternative splicing, missense or nonsense genetic variants or PTMs. In contrast to current gold standard techniques, top-down proteomics permits a more in-depth analysis of protamine PTMs and proteoforms, thereby opening up new perspectives to unravel their impact on male fertility. We report on the analysis of two normozoospermic semen samples by top-down proteomics. We discuss the difficulties encountered with the data analysis and propose solutions as this step is one of the current bottlenecks in top-down proteomics with the bioinformatics tools currently available. Our strategy for the data analysis combines two software packages, ProSight PD (PS) and TopPIC suite (TP), with a clustering algorithm to decipher protamine proteoforms. We identified up to 32 protamine proteoforms at different levels of characterization. This in-depth analysis of the protamine proteoform landscape of normozoospermic individuals represents the first step towards the future study of sperm pathological conditions opening up the potential personalized diagnosis of male infertility.

## 1. Introduction

Protamines are the main nuclear proteins in the sperm cells of many species, and they play a crucial role in the correct packaging of the paternal DNA [1,2,3]. During the final stage of spermatogenesis (called spermiogenesis), the chromatin of immature germ cells undergoes marked remodeling, in which histones are sequentially replaced by specific histone variants, transition proteins and, finally, protamines [1,4,5,6,7]. Both the small size of protamines (20 to 55 amino acids) and their extremely rich sequence in positively charged arginine residues allow the formation of highly condensed toroidal complexes, together with the negatively charged paternal DNA [8,9]. In addition, the high number of cysteine residues in mammalian protamines promotes the generation of multiple intra- and inter-molecular disulfide bonds and zinc bridges between protamines, thereby stabilizing the compact toroidal nucleoprotamine complex [10,11]. Several functions have been proposed for the nucleoprotamine domain, including the protection of the paternal genome during transit from the male to the oocyte and contribution to obtaining the required hydrodynamic shape for mature sperm functionality [3,4,12,13,14,15,16,17]. Of note, the replacement of histones by protamines during spermatogenesis is not complete, resulting in 85–95% of sperm DNA packed by protamines and the remaining 5–15% attached to histones [18]. Several studies have reported that the distribution of the nucleohistone and nucleoprotamine complexes along sperm chromatin is not random. This distinctive chromatin structure has been proposed as a specific epigenetic mark of sperm that could regulate gene expression once the oocyte has been fertilized [19,20,21].

The crucial role of protamines in the compaction of the paternal genome has given rise to considerable interest in the analysis of protamine alterations in the andrology field. Mammalian spermatozoa contain two types of protamines: protamine 1 (P1) and the family of protamine 2 (P2) proteins. P1 is expressed in all species of protamine-containing vertebrates, while the P2 family is present only in some mammalian species, including the human and mouse. P2 is formed by the proteolysis of the P2 precursor (pre-P2), resulting in the HP2, HP3 and HP4 components, of which HP2 is the most abundant [4] (Figure 1).

Several groups have established similar protein levels of P1 and P2 in normal spermatozoa, with an approximate 1:1 proportion, and alterations of the P1/P2 ratio have been associated with male infertility and poor reproductive outcomes [22,23,24]. However, controversial results have been reported in the literature and the P1/P2 ratio shows a wide range in fertile men [25]. The P1/P2 ratio has been commonly assessed by polyacrylamide gel electrophoresis (PAGE) and protein staining [26,27,28,29]. However, this common approach does not provide information on whether a specific alteration of the P1/P2 ratio is due to the deregulation of P1, the P2 family, specific P2 components, or all of three. In addition, the emerging hypothesis that alterations in both histone variants and histone post-translational modifications (PTMs) might lead to male infertility [30] suggests that, in a similar way, alterations in protamine PTMs are also associated with male infertility. Recently, we suggested that the presence of human sperm protamine proteoforms, including truncations or proteoforms containing PTMs, explain the variability observed in the P1/P2 ratio assessed by conventional PAGE [30,31]. Therefore, we propose that more sophisticated methodologies such as mass spectrometry (MS) could help decipher the different protamine proteoforms present in mature sperm cells and their impact on male fertility [31]. However, the peculiar physicochemical properties of protamines (high arginine content with P1 containing 24 Arg residues out of 51 and HP2 32 Arg residues out of 54) and the high similarity between P2 components make them unsuitable for characterization by bottom-up proteomic strategies. The main reason is that conventional trypsin digestion results in extremely small protamine peptides that are lost during their processing. Additionally, the intra- and inter-protamine disulfide bonds and zinc bridges that form due to the high number of cysteine residues in the protamine sequences (6 Cys in P1 and 5 Cys in HP2) require methods for reduction and alkylation to avoid multiple cysteine oxidations. Of note, the low molecular weight of protamines (P1: 6687 Da and P2: 12,912 Da) is a big advantage for analyzing these molecules by top-down proteomics.

Top-down proteomics is a powerful MS-based technology that allows direct analysis of all proteoforms, whether derived from alternative splicing, missense or nonsense genetic variants or PTMs. This approach offers unique and precise information on proteoforms that is complementary to that achieved using bottom-up proteomics. The top-down analysis of MS-derived data is challenging both due to the deconvolution of the spectra and subsequent searches in databases for correct proteoform assignment. Unlike bottom-up proteomics, where robust solutions are found for the identification, characterization and quantification of peptides and proteins, data analysis in top-down proteomics is not yet fully established. In this regard, it is still one of the bottlenecks for top-down proteomic strategies. One of the main pitfalls is the lack of automated validation of the final IDs, which requires manual inspection of the assignments.

There are two main approaches for top-down data analysis and proteoform identification: (a) analysis based on a search in annotated databases, where all the possible proteoforms with defined PTMs are included in a certain position, like in ProSight PD (PS) [32], and (b) analysis based on spectral alignment algorithms [33], which matches the tandem mass spectra of the proteins with the unannotated database, without prior knowledge of the type of PTMs or their location, like in TopPIC suite (TP) [34,35].

The data analysis workflow that we describe herein combines two distinct software packages (referred to herein as nodes), namely PS and TP. To avoid redundant identifications on the same precursor, our pipeline is based on MS1 clustering and harmonization of the two datasets obtained. This new strategy allowed us to identify 13 proteoforms of P1 and 19 proteoforms of P2 in human semen samples. We envisioned that this strategy could be applied to other highly modified proteins such as histones.

## 2. Materials and Methods

### 2.1. Sample Collection

Human semen samples (*n* = 2) were obtained from patients undergoing routine semen analysis at the Assisted Reproduction Unit at the Clinic Institute of Gynecology, Obstetrics and Neonatology, at the Hospital Clínic of Barcelona, Spain. All patients gave signed informed consent in accordance with the Declaration of Helsinki. The ejaculates were collected by masturbation into sterile containers after 3–5 days of sexual abstinence. Seminal parameters were evaluated using the automatic semen analysis system CASA (Proiser, Paterna, Spain). After seminal analysis, all samples were classified as normozoospermic according to World Health Organization guidelines [36]. Subsequently, sperm was purified through 50% Puresperm^®^ density gradient separation (NidaCon International AB, Gothenburg, Sweden), following the manufacturer’s instructions.

### 2.2. Purification of Human Sperm Protamine

The protamine-enriched fraction of spermatozoa was purified as described in [27]. except that histones and other basic proteins weakly associated with DNA were removed from the sperm by 0.5 M HCl, vortexing and incubation for 10 min at 37 °C and centrifugation at 2000× *g* for 20 min at 4 °C (three times) prior to suspending the sperm pellet with 0.5% Triton X-100, 20 mM (pH8) Trizma^®^ base HCl, and 2 mM MgCl_2_. Samples were then processed as described in [27]. For the top-down MS approach, protamine-enriched fractions (equivalent to 15 million spermatozoa) were dried at room temperature overnight under a fume hood and kept at 20 °C. Before liquid chromatography MS analysis (LC-MS/MS), samples were reconstituted in 50 mM NH_4_HCO_3_, reduced with 10 mM dithiothreitol (DTT) for 45 min at 56 °C, alkylated for 30 min in the dark with 50 mM iodoacetamide (IAA) and further desalted using PolyLC C18 filter tips (PolyLC INC., Columbia, MD, USA).

### 2.3. Protamine Quantification

Purified extracts of protamines were quantified as described in [27]. Briefly, an aliquot of protamines for each sample was separated in an acid-urea PAGE together with increasing amounts of a protamine standard obtained as described elsewhere [37]. Approximate amounts of P1 and P2 were calculated from protamine optical density measurements and the quantification obtained with the protamine standard, using Quantity One 1-D Analysis Software (Bio-rad, Hercules, CA, USA).

### 2.4. Top-Down Proteomics

#### 2.4.1. LC-MS

We analyzed two biological replicates, each with two technical replicates. Samples were reconstituted in an aqueous solution of 3% acetonitrile (ACN) and 1% formic acid (FA) and injected to the LC-MS/MS system.

LC-MS/MS analysis of intact protamines was conducted on a Dionex Ultimate 3000 coupled to an Orbitrap Fusion Lumos mass spectrometer (both from Thermo Scientific, San Jose, CA, USA). Samples were loaded onto C18 trap columns (100 µm × 2 cm Acclaim PepMap100, 5 µm, 100 Å (Thermo Scientific) at a flow rate of 15 µL/min. Protein content was separated using C18 analytical columns (Acclaim PepMap^®^ RSLC 75 µm × 50 cm, nanoViper, C18, 2 µm, 100Å (Thermo Scientific) and a linear gradient from 3 to 15% B in 30 min at a flow rate of 250 nl/min (A = 0.1% FA in water, B = 0.1% FA in ACN). The Advion TriVersa NanoMate (Advion Ltd., Harlow, United Kingdom) was used as the nanospray interface with the direct on-line coupling mode with a spray voltage of 1.60 kV. The mass spectrometer was operated in data-dependent acquisition (DDA) mode. Survey MS scans were acquired in the orbitrap with the resolution (defined at 200 *m*/*z*) set at 120k. The top N (most intense) ions per scan were fragmented by electron transfer dissociation (ETD) and detected in the orbitrap at 120k. The ion count target value was 400,000 for the survey scan and 1,000,000 for the ETD-fragmented MS/MS scan. Target ions already selected for MS/MS were dynamically excluded for 30 s. RF lens was tuned to 30%. The minimal signal required to trigger the MS to MS/MS switch was set at 500,000.

#### 2.4.2. Proteoform Search

We devised a twin search approach for our top-down proteomics workflow (Figure 2). This strategy encompasses the bioinformatic stools TP and PS.

The first node used for top-down searches was TP v1.4.2, including its tool TopFD to deconvolute the spectra. This node can identify proteoforms of an entire proteome with remarkable speed by searching against a FASTA database without prior knowledge of possible modifications or existing PTMs. Therefore, TP is particularly suitable when dealing with highly modified proteins such as histones or protamines.

We included the following PTMs in the parameters of the TP search: oxidation (in M), phosphorylation (in STY), acetylation (in KS), methylation, dimethylation (both in KR) and trimethylation (in K) for being the major PTMs found in histones and therefore also expected in protamines. We included the following N-terminal forms: none, N-terminal methionine excision (NME), NME_Acetylation and M_Acetylation, which referred to proteoforms with or without the initial methionine non-acetylated or acetylated. We also defined precursor and fragment mass tolerances of 15 ppm, and maximum mass shifts of 500 Da. We only considered Proteoform Spectrum Matches (PrSMs) with FDR < 1% for downstream data integration. We used a FASTA file containing P1 and P2 sequences as a database.

The second search was done with Proteome Discoverer v2.5.0.400 (Thermo Fisher Scientific) and its PS v4.0.0.228 node. In contrast with TP, PS requires a database with all theoretical proteoforms explicitly listed. The Protein Annotation option from the PS Database Manager allows such a database to be built from a typical *.xml file available in SwissProt.

It is crucial to mention here the existing bioinformatic bottleneck despite current computing power. As the number of PTMs for a given protein sequence increases, the number of entries in the database rapidly explodes. We can compute the number of theoretical proteoforms for a given database, *N*, using the following expression:(1)$$N=\overset{P}{\underset{j=1}{\sum }}\left[\overset{n}{\underset{i=1}{\prod }}{\left(m_{i}+1\right)}_{j}\right]$$
where *P* is the number of protein sequences in the database, *n* the number of amino acids for each sequence, and *m* the number of possible modifications for each amino acid on each sequence [38]. Given the PTMs typically found in protamines that we previously mentioned, this equation returns around *N*_P1_ ≈ 4.4 × 10^15^ and *N*_P2_ ≈ 3.9 × 10^22^ theoretical proteoforms for P1 and P2, respectively, which gives *N* ≈ *N*_P2_ ≈ 10^22^ in total. These values would imply an extremely long computational search time. To overcome this issue, we launched a preliminary PS search excluding those PTMs on consecutively repeated arginine residues, but keeping PTMs in selected arginines from the N-terminal, C-terminal and from the middle of the sequence. As a result, the number of theoretical proteoforms in this first search was reduced by 15 orders of magnitude. Since this preliminary search did not find any acetylations, methylations, dimethylations or trimethylations, we launched a second three-tier search (Annotated Proteoform, Subsequent Search and Annotated Proteoform) using a database with only phosphorylations (in STY), oxidations (in M) and acetylation in N-terminal in all the nodes. Using this strategy, we further reduced the number of theoretical proteoforms by 2 orders of magnitude, adding up a total of 17 orders of magnitude and thereby giving a much more reasonable search time. Precursor mass tolerances were 2.2 Da/10 ppm/500 Da for Annotated Proteoform, Subsequent Search and Annotated Proteoformrespectively, and fragment mass tolerances 10 ppm in all cases. The mass spectrometry proteomics data have been deposited to the ProteomeXchange Consortium via the PRIDE [39]. partner repository with the dataset identifier PXD024405.

#### 2.4.3. Data Analysis

The PrSM output format used by PS and TP is highly heterogeneous. The column name used by each node to designate the same concept usually varies (for example: “Spectrum File” in PS and “Data file name” in TP). Even the unit used to express the same magnitude differs (for example: TP expresses retention time in minutes while PS uses seconds). We computationally addressed this issue with a custom Python code [40]. (available on the GitHub repository: https://github.com/MSPCF/2021_Proteomes, accessed on 27 April 2021). To this end, we leveraged some of Python’s most popular packages such as Pandas [41], NumPy [42], Biopython [43], Plotnine [44], and Seaborn [45].

Once this harmonized dataset with all PrSMs was ready, we filtered out those identifications with retention times beyond 42 min for being associated with the column wash. We did not apply any PrSMs pre-filter based on PrSM quality (such as the MS1 intensity or the E-value), but this might be convenient depending on the overall PrSM dataset quality. After pre-filtering, we performed the PrSM clustering by means of a density-based spatial clustering algorithm of applications with noise (DBSCAN) [46,47]. implemented within the Scikit-learn Python package [48]. This clustering algorithm is particularly suitable in our scenario because it does not require pre-definition of the number of clusters expected (a priori, we do not know the number of proteoforms within a sample). In addition, DBSCAN has just two main hyperparameters to work with: maximum distance between two samples to accept both as belonging to the same neighborhood (*ε*), and minimum number of samples in a neighborhood to be accepted as an independent cluster (*n*_min_). In our scenario, “samples” and “independent clusters” would correspond to PrSMs and proteoforms, respectively. Regarding the “distance”, it is important to mention that we simply passed the standard score (commonly called z-score) of the deconvoluted PrSM masses to the clustering algorithm. Finally, we optimized the DBSCAN hyperparameters *ε* and *n*_min_ by interactive exploratory data analysis with a custom Python code leveraging the Bokeh Python package [49], getting *ε* = 1.2 × 10^−3^ (which in our filtered harmonized dataset corresponds to 1.28 Da) and *n*_min_ = 10.

#### 2.4.4. Proteoform Annotation

Once clustering was done, proteoforms were annotated in a semi-automatic manner comprising two major steps. For a given cluster, we first checked to see if most of its PrSMs pointed to a particular protein, usually any full-length or truncated mature or immature protamine form or other sperm-related protein. In this case, we pre-assigned what we call a “cluster-protein” to this cluster and then computed its corresponding cluster-protein mass shift as follows:(2)$$\Delta m_{CT}=m_{C}-m_{T},$$
where *m_T_* is the theoretical mass of the fully carbamidomethylated cluster-protein, and *m_C_* the mean mass of all PrSMs within this cluster. When a predominant cluster-protein was not found, we manually attempted to assign a suitable protein consistent with the particular *m_C_* value. Secondly, and depending on the precise Δ*m*_CT_ we manually assigned a suitable PTM set. The proteoform annotation table is provided in the Appendix A, as well as the associated harmonized PrSMs dataset, including the clustering outcome and the proteform annotation (Appendix A).

## 3. Results and Discussion

We analyzed two biological replicates (A and B), with two technical replicates each (R01 and R02 for A, R03 and R04 for B). The results presented here are all the data combined with the final aim of characterizing the protamine proteoforms in the samples.

In the following sections we will use the five-level classification system for proteoform identifications proposed by Smith et al. [50]. We will refer to levels 1, 2A, 2B and 3. Level 1 describes proteoforms identified with full knowledge of their gene origin, amino acid sequence and, when present, the identification and localization of their PTMs. Level 2A refers to those proteoforms with a defined amino acid sequence from a known gene with all the PTMs identified but with incomplete localization. In level 2B, the PTMs are not fully characterized but their localization is complete. Level 3 refers to those proteoforms that are certain for two of the four ambiguity classes: gene, sequence, PTM identification and PTM localization. In this manuscript we refer to level 3 as proteoforms with a gene and sequence defined but with an unknown modification and localization.

### 3.1. Overview of the DBSCAN Approach

The goal of our DBSCAN approach was to integrate and harmonize the PS and TP datasets by PrSM deconvoluted experimental masses. We assumed that a proteoform at level 1 without PTMs, or level 2A or level 3 with an equivalent mass shift, always had the same mass and therefore could be annotated as the same proteoform. In this first step, we were not able to localize modification sites and reach level 1 for proteoforms with PTMs, but we localized PTMs in a second step considering only those PrSMs identified with a high confidence level (Section 3.4). The power of our workflow for top-down proteomic data analysis lies in combining multiple nodes to achieve high-quality PrSMs and a single harmonized dataset.

From the PS and TP searches, a total of 2838 PrSMs passed our filters (56.6% for PS and 43.4% for TP) and were further submitted for clustering. The DBSCAN algorithm successfully clustered 2578 of the PrSMs (55.9% for PS and 44.1% for TP) into 27 well differentiated mass clusters, leaving 260 unclustered PrSMs (63.8% for PS and 36.2% for TP).

We identified the following four main protamine groups: P1 proteoforms, comprising clusters from P|000 to P|008 (7029 Da to 7311 Da); mature P2 forms (HP2, 3 and 4) from P|009 to P|013 (7538 Da to 8123 Da); intermediate P2 forms (HPS1, 2 and HPI2) from P|014 to P|020 (9002 Da to 10,654 Da); and pre-P2 forms from P|021 to P|025 (13,195 Da to 13,516 Da) (Table 1). Cluster P|026 remained unidentified and could presumably be ascribed to a sperm-related protein other than protamines. In addition to several phosphorylated proteoforms (up to +80 Da × 4), we also identified some presenting an unrecognized +61 Da mass shift modification (up to +61 Da × 3), usually in combination with other P1 phosphorylated proteoforms. We found this +61 Da mass shift in P1, HP2, HP3, HP4, HPS1 and HPS2. We had already identified this +61 Da mass shift in previous studies [31]. We cannot explain this +61 Da mass shift yet but propose that these protamines exhibit some metal ion-binding sites for Zn2+ or Cu+ [51,52]. Further verification is required using other MS-based techniques such as inductively coupled plasma MS. Finally, we also observed the formation of the cyclic pyroglutamyl residue (pyro-Glu) from the N-terminal glutamine of HPS1 [53].

We used different projections to visualize the clustering outcome and better optimize its *ε* and *n*_min_ parameters with our Bokeh-based web application. In particular, we inspected the following three projections: retention time vs. deconvoluted experimental mass (Figure 3a, Appendix A), log_2_ intensity vs. deconvoluted experimental mass (Appendix A), and retention time vs. log_2_ intensity (Appendix A). This strategy was crucial for gaining greater insight into how the DBSCAN algorithm worked, as well as the differences between the PS and TP nodes.

The scatterplot of the retention time vs. the deconvoluted experimental masses was particularly useful. We used colors to differentiate clusters. In this plot, it is easy to see proteoforms with the same mass clustered or to localize PrSMs that did not cluster. The four main protamine groups mentioned above per mass region can be visualized in the same plot, grouped in mass regions: (a) P1: 7000 to 7500 Da; (b) mature P2: 7500 to 8200 Da; (c) Intermediate P2: from 9000 to 11,000 Da; and (c) pre-P2: from 13,000 to 13,600 Da. Each region had the canonical sequences and the subsequent proteoforms with different modifications.

Using the DBSCAN approach, we identified proteoforms at levels 1, 2A and 3 of the classification system [51]. We reached level 1 in proteoforms without PTMs (truncated and non-truncated forms) and in the case of HPS1 +PyroGlu, level 2A in proteoforms with known modifications (phosphorylations), and level 3 with proteoforms with the +61 mass shift.

### 3.2. Advantages of the DBSCAN Approach vs. PS and TP

When handling PS and TP outputs, we encountered some common difficulties and issues, such as the different output tables generated or incorrectly assigned proteoforms. With the DBSCAN approach, we overcame some of these issues and integrated, simplified and corrected the PS and TP outputs.

We first examined the results at the amino acid sequence level without considering known or unknown modifications. TP identified 65 sequences, of which 41 were truncated at the N-terminal, 5 at the C-terminal and 4 in both the N and C-terminal regions. PS found 14 sequences, of which 9 were truncated at the N-terminus (Appendix A). These numbers, based only on the amino acid sequence level, already highlighted the high dispersion between the nodes. When we introduced modifications, the dispersion between results was even higher, thereby hindering comparison of the two datasets. However, after the cluster analysis using the PS and TP data, most of these sequences fell into the 8 sequences previously described in Uniprot (Figure 1).

Going deeper into the data analysis, we observed that PS and TP sometimes failed to assign a proteoform from a given PrSM. We identified two main common cases of incorrect PrSM assignment: (a) PrSMs with very different deconvoluted masses recognized as the same proteoform (which should not be possible for any classification level), and (b) PrSMs with a very similar deconvoluted mass recognized as different proteoforms. The first misassignment case (from now on “converging”) was usually found in PS while the second one (from now on “diverging”) was found in both PS and TP. Figure 3 illustrates these two misassignment scenarios by zooming in to highlight the mass region where HPS1 and HPS2 are located (Figure 3b). For example, PS considers those PrSMs around 9000 Da and 9060 Da to be the same proteoform (Figure 3b, green data points). Another example, TP does not report those PrSMs clustered around 9283 Da as the same proteoform (Figure 3b, purple and magenta data points). Note how our DBSCAN approach nicely groups those PrSMs with similar masses into well-defined clusters (proteoforms), in striking contrast to how PS and TP perform (Figure 3c).

#### 3.2.1. Converging and Diverging Misassignment of PrSMs

We considered converging misassignment to be those cases in which PrSMs with very different experimental deconvoluted masses (i.e., belonging to different proteoforms according to our clustering approach) are identified by the node as the same proteoform. In general, for a given TP proteoform ID we found just one proteoform according to our clustering approach, meaning that TP did not mix proteoforms with different deconvoluted masses under the same proteoform ID. On the contrary, we observed that PS “converges” into the same proteoform ID PrSMs belonging to different proteoforms according to our clustering approach. In fact, PS classified under the same proteoform ID PrSMs with deconvoluted experimental mass differences up to 980 Da. This PS issue usually occurred when there were no identified modifications, as in the +61 mass shift case.

A clear example of this situation can be observed in Figure 3b. Note how PS converged as HPS2 (theoretical mass 9002.75 Da) PrSMs with quite different deconvoluted masses (9002 Da and 9064 Da). We consider that PrSMs around 9003 Da (cluster P|015) are correctly associated with HPS2, but those around 9064 Da (cluster P|016) should be linked to a new HPS2 +61 Da proteoform because they are clustered apart (Table 2a). We manually validated this hypothesis with the help of ProSight Lite [54], by precisely localizing such +61 Da mass shifts in the same spectrum identified as HPS2 by PS (Table 2b). Note how both the P-Score and the number of matching fragments improved once we assigned this + 61 Da mass shift to the C-46 residue in HPS2 (the first decreases from 1.1 × 10^−91^ to 7.6 × 10^−155^ and the second increases from 62 to 95) (Table 2a,b). Similarly, PS also “converged” as HPS1 (9300.91 Da), PrSMs with clearly different deconvoluted masses (this time with values around 9301 Da and 9362 Da). Again, since those PrSMs were clustered apart (P|019 and P|020, respectively), we assumed that PrSMs around 9362 Da correspond to a new HPS1 +61 Da proteoform (Figure 3b).

We considered diverging misassignment to be those cases in which PrSMs with very similar experimental deconvoluted masses (i.e., belonging to the same proteoform according to our clustering approach) were identified as different proteoforms. Both PS and TP tended to make this kind of mistake. Diverging misassignment translated into the identification of multiple proteoforms (according PS or TP) within one of our mass clusters. In particular, the average number of proteoforms per cluster according to PS and TP was 2.81 and 2.96, with extreme cases in which we found up to 9 and 5 proteoforms per cluster, respectively.

A clear example of this situation can be seen in Figure 3b (for PS and TP nodes) and Table 1. Note how the PS and TP nodes “diverged” those PrSMs with deconvoluted experimental mass around 9283 Da (cluster P|018) in up to nine and three distinct proteoforms, respectively. We consider that all these PrSMs should be assigned to an HPS1 proteoform with a pyroglutamic acid at the glutamine located in its N-terminus (−18.010565 Da). Similarly, as in the previous Section 3.2.1, we manually validated our hypothesis using ProSight Lite by including and localizing this pyroglutamic acid (Table 2d–f). Note how for two particular PrSMs, which PS originally assigned to HPS1 + 2 phosphorylations (Table 2c) and HPS2 + 1 phosphorylation + 1 N-term acetylation (Table 2e), both the P-Score and the number of matching fragments improved once we localized the pyroglutamic acid at HPS1 N-terminus (Table 2d,f) (the first decreases from 7.9 × 10^-38^ to 2.8 × 10^−103^ and from 9.7 × 10^−45^ to 1.8 × 10^−118^, respectively, and the second increases from 35 to 75 and from 40 to 85, respectively).

#### 3.2.2. DBSCAN Approach Limitations

The DBSCAN approach was able to merge, harmonize and correct the PS and TP results but had some limitations. The three main limitations were (a) the requirement for previous knowledge of the sample to perform the final semi-manual annotation; (b) the assumption that the same mass meant the same proteoform when two amino acid sequences (with or without PTMs) may in fact have the same mass; and (c) some of the identified PrSMs remained unclustered and were discarded.

At the present stage of development, proteoform assignment to each cluster is semi-automatic and requires a minimum degree of manual validation and previous knowledge of the sample. We therefore believe that this approach can easily be applied to simple samples even when they have a large number of modifications, such as protamines or histones. However, we still have to check its viability in complex samples.

One of the strengths of the DBSCAN approach was also a limitation. We assumed that one mass corresponded to one proteoform without the modification localized. This assumption corrected and simplified the data, but could also lead to incorrect assignments. It is possible for two different proteoforms to have the same mass even if they have different amino acid sequences. To address this limitation and try to distinguish this kind of situation, we intend to add the retention time as a new parameter in the DBSCAN algorithm in a future version of the workflow. This modification would better differentiate and monitor proteforms of the same mass.

As stated in Section 3.1, the DBSCAN algorithm successfully clustered 2838 of the PrSMs, leaving 260 unclustered. Moreover, only one (P|026)—comprising an additional 28 PrSMs—out of 27 clusters was unknown. As a result, our approach left a total of 288 (10.1%) unclassified PrSMs (unclustered/unidentified). However, we observed that unclassified PrSMs showed overall poorer quality than classified ones (clustered/identified).

In particular, unclassified PrSMs had a lower mean E-value and intensity than classified PrSMs for both PS and TP (Appendix A). The mean of the -log_10_ of the E-value was 73.4 (PS) and 26.6 (TP) for the classified PrSMs and 32.3 (PS) and 11.1 (TP) for the unclassified ones. The mean of z-score normalized log_2_ intensities was 0.10 (PS) and 0.18 (TP) for the clustered PrSMs and −0.77 (PS) and −1.91 (TP) for the unclustered ones. The loss of 10.1% of all PrSMs is therefore not particularly concerning since it implies that low confidence matches are discarded.

The DBSCAN approach combines the strengths of the PS and TP nodes, in particular the high spectra identification rate of PS and the ability to identify proteoforms with an arbitrary mass shift of TP. The clustering approach also overcomes the weaknesses of the two nodes, like the two kinds of proteoform misassignment highlighted in Section 3.2.1.

### 3.3. Protamine Proteoform Landscape and Protamine Ratios

Our DBSCAN-based approach allowed us to identify a total of 26 protamine proteoforms. The protamine forms involved in this proteoform landscape were P1, pre-P2, HPI2, HPS1, HPS2, HP2, HP3 and HP4. Among the PTMs contributing to the landscape, we found phosphorylation, N-terminal cyclic pyroglutamyl, N-terminal glutamine loss and the +61 Da modification, which was discussed in Section 3.1.

To gain greater insight into this rich protamine proteoform landscape, we performed scatterplots for the cluster-protein mass shift associated with each protamine proteoform cluster (Δ*m_CT_* in Equation (2)) vs. the corresponding protamine forms so that each data point represents a different 1, 2A- or 3-level proteoform (Figure 4). We generated this scatterplot at three levels: global, by technical replicate, and by biological replicate (Figure 4, panels a, b and c, respectively).

Note how 22 out of 25 proteoforms appearing in biological replicate A are found within its two technical replicates R01 and R02 (exceptions are P1 +1ph +2(61), pre-P2 +4ph and HP3 +1(61)). Similarly, 24 out of 26 proteoforms appearing in biological replicate B are found within its two technical replicates R03 and R04 (exceptions are P1 +2ph +2(61) and HP3 +1(61)). The only proteoform not found in the two biological replicates was P1 +2ph +2(61). By revisiting Table 1, we checked that all these exceptions corresponded to clusters with limited PrSM counts. In particular: P1 +1ph +2(61) with 20 + 29 (TP + PS) counts; pre-P2 +4ph with 6 + 8; HP3 +1(61) with 13 + 17; and P1 +2ph +2(61) with 5 + 7. After inspecting the protamine proteoform landscapes at all levels, we saw that the performance of our characterization was acceptable for intertechnical replicate variability.

As an advantage over classical protamine ratio assessment through gel-staining approaches, our strategy enables computation of the abundance ratio between any protamine proteoforms. The ratio between P1 and P2 mature proteoforms (HP2, HP3, and HP4) is the ratio most widely used for the study of male infertility [3]. Alterations of the P2 family seem to be more frequent than those of P1, and the incomplete processing of the pre-P2 has also been linked to male infertility [55,56,57]. Therefore, the ratio of the different protamine proteoforms between immature proteoforms of P2, including pre-P2 and intermediate proteoforms (HPI2, HPS1, HPS2) and mature forms of P2, is highly relevant in the context of understanding the chromatin maturity state. We provided barplots for these two ratios (P1/P2-mature and P2-immature/P2-mature) at three levels: global, by technical replicate, and by biological replicate (Figure 4, panels d, e and f, respectively). To better monitor the contribution of each node in each ratio computation, we split them using a color scale. The P1/P2-mature and P2-immature/P2-mature ratios fall around 1 for our two biological replicates and their corresponding technical replicates. Note how PS gives slightly greater ratios than TP. However, this systematic discrepancy is smaller than associated standard deviations.

### 3.4. Modification Site Assignment

The DBSCAN cluster approach allowed us to identify modified proteoforms at level 2A or 3. We reached level 2A with phosphorylated proteoforms, but we did not localize the p-site. We reached level 2B classification for HPS2 +61 Da shown in Table 2c. However, this localization was done manually. Undertaking this task for all PrSMs would have been time-consuming, so we accepted that proteoforms modified with +61 Da were identified at level 3.

To reach level 1 of proteoform identification for phosphorylated proteoforms, we analyzed those modified PrSMs according to PS and TP. We took into account PrSMs with a 2A classification level that matched our cluster assignment and filtered those with a high localization confidence level (PTM localization probability > 75% in TP and C-Score > 40 in PS [58].). A total of 111 p-sites passed the filter (33% of the total), 63 for PS and 48 for TP. We manually checked two spectra for P1 +ph and P2 +2ph and their localization sites to be confident of the automatic assignment by the softwares (Figure 5, panel a). Both nodes found P1 and pre-P2 phosphorylated (P1 monophosphorylated and pre-P2 with one, two, three and four p-sites). PS and TP mainly localized the phosphorylation of P1 in site S-11 with 20 and 23 PrSMs, respectively. TP also found PrSMs with the phosphorylation site in S-13, Y-16, S-22 and Y-44. While S-13 was identified with 11 PrSMs, the other three sites were identified with only one PrSM each. However, PS and TP showed a distinct behavior when localizing p-sites in pre-P2. TP found pre-P2 monophosphorylated in S-59, S-13 and S-51 and PS found pre-P2 bi-phosphorylated in S-8 and S-10, tri-phosphorylated in Y-4, S-8 and S-10 and tetra-phosphorylated in Y-4, S-8, S-10 and T-98 (Figure 5, panel b). Given the above observations, we concluded that TP tends to localize mono-phosphorylations while PS performs better at localizing multiple sites.

Protamine phosphorylation sites have been studied in humans and other mammalian species through conventional approaches [15,59,60,61,62]. For instance, protamine 1 phosphorylation at S11 has been described in humans and is conserved among other species, as well as at S9 and S29 residues [24,63,64,65]. Regarding pre-P2, phosphorylation sites have been identified at S8, S10, S37, T47, S51, S59 and S73 [24,63,64,65]. Our results not only are consistent with these previous evidences, but also have allowed the identification of novel proteoforms in the human sperm [31]. These results support this new DBSCAN-based strategy combining PS and TP as a robust approach for identifying site-specific mono- and poly-phosphorylated human protamines.

## 4. Conclusions

Our clustering approach combined the strengths, and overcame the weaknesses, of two popular top-down proteomics bioinformatics tools: ProSight PD (PS) and TopPIC Suite (TP). We showed the usefulness of the DBSCAN algorithm to solve some of the misassignments found when analyzing top-down database searches provided by these two nodes. From a qualitative viewpoint, and despite the limited number of biological replicates available (*n* = 2), our proteoform characterization showed good reproducibility across individuals since only a level 3 proteoform was not identified in both biological replicates, namely, P1 +2ph +2(61). With this strategy, we identified 13 proteoforms of P1 and 19 proteoforms of P2, 11 of them with the p-site localized. Our results highlighted the capacity of our clustering approach to characterize purified small proteins with a high number of truncated forms or PTMs. The possibility of examining alterations through studying individual protamine proteoforms, including the assessment of specific modified proteoforms, is a first step towards a personalized diagnosis.

## Figures and Tables

**Figure 1 proteomes-09-00021-f001:**
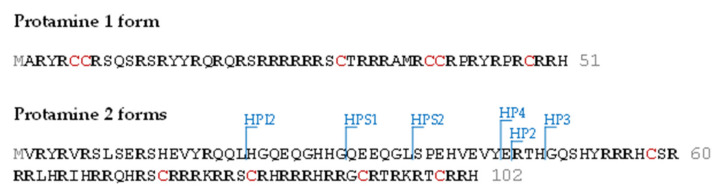
Human protamine amino acid sequences. Protamine 1 is translated as a mature form containing 51 residues (P1). In contrast, protamine 2 is translated as a precursor protein containing 102 residues (pre-P2), which is processed by proteolysis in multiple intermediate forms (HPI2 (22–102), HPS1 (34–102) and HPS2 (37–102)) and mature forms (HP2 (46–102), HP3 (49–102) and HP4 (45–102)).

**Figure 2 proteomes-09-00021-f002:**
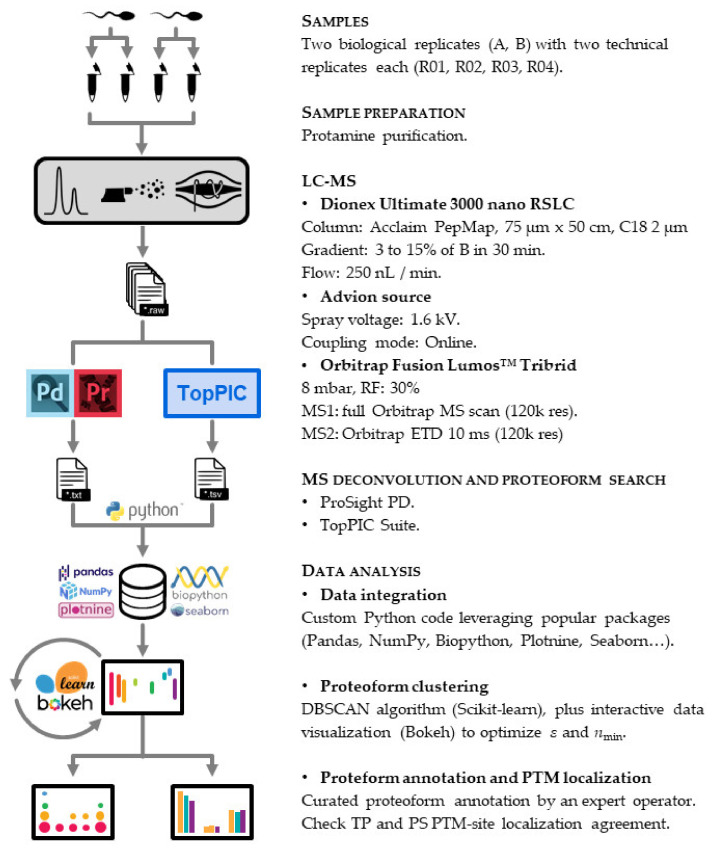
Protamine characterization workflow. We purified the protamine content of two biological replicates (A, B) and injected four technical replicates (R01, R02, R03, R04) into the LC-MS instrument. We then used two bioinformatic tools (ProSight PD (Thermo Scientific, San Jose, CA, USA) and TopPIC Suite (Indianapolis, IN, USA)) to identify proteoform spectrum matches. The whole data analysis pipeline downstream was implemented entirely using Python as a programming language.

**Figure 3 proteomes-09-00021-f003:**
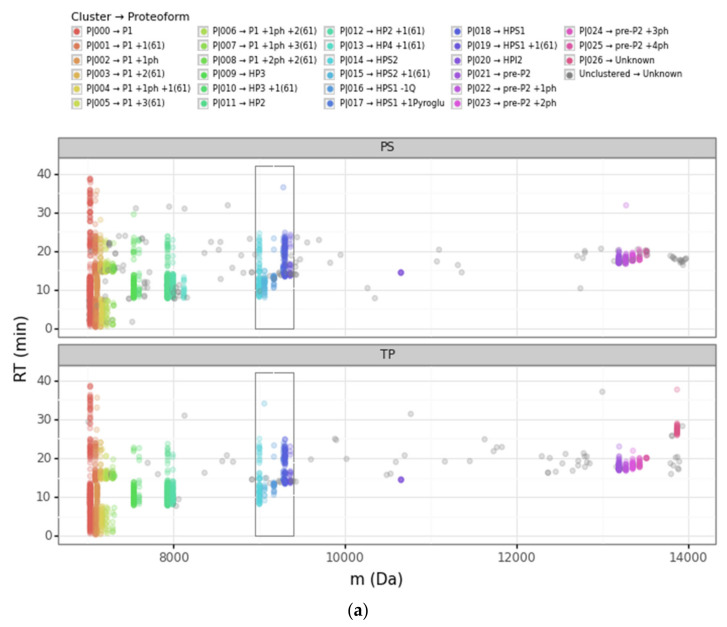
Scatterplots of the retention time (RT) vs. the deconvoluted experimental mass (m) for PS (Prosight PD) and TP (TopPIC Suite). (**a**) Deconvoluted masses from 6000 Da to 14,000 Da. The sequential color scale highlights the clusters obtained sorted by increasing experimental mass, while unclustered PrSMs appear in grey. (**b**,**c**) Deconvoluted masses from 8900 Da to 9400 Da corresponding to clusters that we associated with HPS1 and HPS2 proteoforms (the region highlighted with a rectangle in panel (**a**)). The color scale in panel (**b**) denotes each proteoform according to PS and TP, while the color scale in panel (**c**) denotes each proteoform according to our clustering approach. (ph = phosphorylation, Q = glutamine, pyroGlu = pyroglutamic acid).

**Figure 4 proteomes-09-00021-f004:**
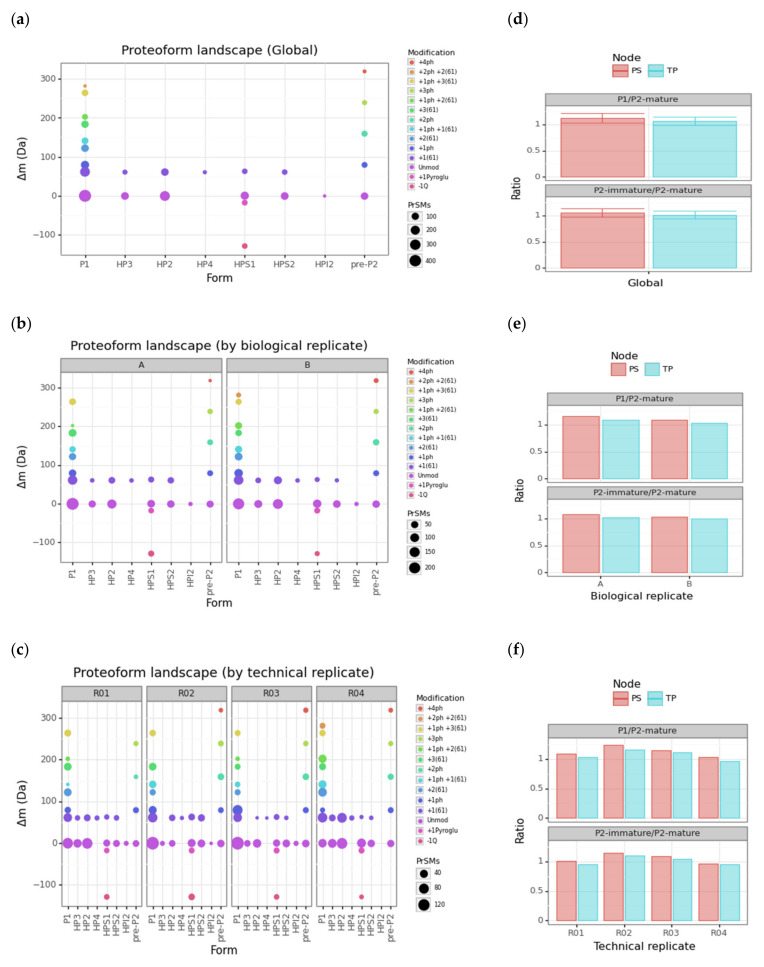
Protamine proteoform landscapes and ratios. (**a**–**c**) Scatterplot of the mass shift vs. the protamine form at global, biological replicate and technical replicate levels, respectively. The sequential color scale in these three scatterplots highlights the particular set of PTMs. The data point size is proportional to the number of PrSMs hitting the corresponding protamine proteoform. (**d**–**f**) Barplots with P1/P2-mature and P2-immature/P2-mature protamine ratios according to each node at global, biological replicate and technical replicate levels, respectively. Error bars correspond to standard deviations of associated technical replicates (ph = phosphorylation, Q = glutamine, pyroGlu = pyroglutamic acid).

**Figure 5 proteomes-09-00021-f005:**
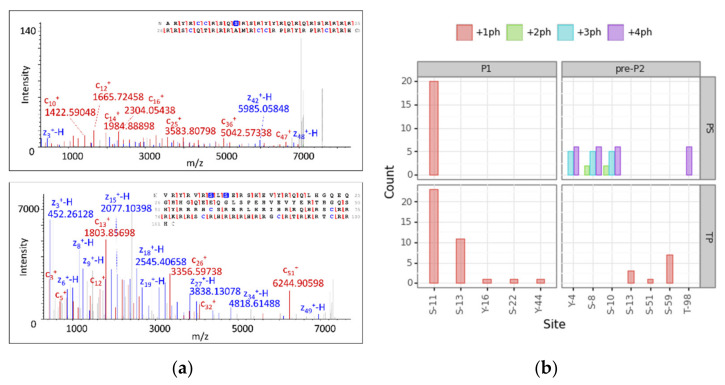
Protamine proteoform phosphosite localization analysis. (**a**) MS2 deconvoluted spectra and their fragment map of P1 +ph (up) and P2 +2ph (down) showing all detected c and z ions and p-site localizations (phosphorylated amino acids are marked in blue). (**b**) Barplots report the number of PrSMs hitting the corresponding localization site. The faceting grid breaks down the phosphosite count for P1 and pre-P2 protamine forms (left and right), as well as for each node (up and down). Color scale denotes the number of phosphorylations coexisting in the same sequence (ph = phosphorylation).

**Table 1 proteomes-09-00021-t001:** Cluster annotation summary. We provide the cluster-to-proteoform correspondence in columns “Cluster” and “Proteoform”. Column “*m_T_* (Da)” gives the theoretical mass of the fully carbamidomethylated cluster-protein (for HP4 the mass is 8062.32 Da). Columns “*m_C_* (Da) (mean)” and “*m_C_* (Da) (std)” give the mean mass and the mass standard deviation of all PrSMs comprising each cluster. Column “Δ*m_CT_* (Da)” gives the cluster-protein mass shift returned by Equation (2). We also report proteoform and PrSM counts according to PS and TP nodes in columns “PF count” and “PrSM count”, respectively (ph = phosphorylation, Q = glutamine, pyroGlu = pyroglutamic acid, PF = proteoform, PrSMs = proteoform spectrum matches, ProSight = PS, TopPIC = TP).

Cluster	Proteoform	*m_T_* (Da)	*m_C_* (Da)(mean)	*m_C_* (Da)(std)	Δ*m_CT_* (Da)	PF Count		PrSM Count	
						PS	TP	PS	TP
P|000	P1	7029.6	7029.45	1.39	−0.14	2	5	247	206
P|001	P1 + 1(61)	-	7090.92	1.92	61.32	5	4	156	97
P|002	P1 + 1ph	-	7108.86	1.58	79.27	3	5	71	74
P|003	P1 + 2(61)	-	7151.59	1.56	122	6	3	71	46
P|004	P1 + 1ph + 1(61)	-	7170.39	1.67	140.79	2	3	51	32
P|005	P1 + 3(61)	-	7212.91	1.85	183.32	4	3	64	41
P|006	P1 + 1ph + 2(61)	-	7231.82	1.66	202.23	6	2	29	20
P|007	P1 + 1ph + 3(61)	-	7293.52	2.73	263.92	4	5	42	29
P|008	P1 + 2ph + 2(61)	-	7310.92	1.43	281.32	3	2	7	5
P|009	HP3	7539.07	7538.54	1.26	−0.53	2	2	72	53
P|010	HP3 + 1(61)	-	7599.63	1.20	60.56	1	2	17	13
P|011	HP2	7933.28	7932.73	1.08	−0.55	1	4	142	119
P|012	HP2 + 1(61)	-	7994.03	1.35	60.75	1	1	64	51
P|013	HP4 + 1(61)	-	8122.66	0.49	60.34	2		16	
P|014	HPS2	9002.75	9002.09	1.10	−0.66	2	3	69	54
P|015	HPS2 + 1(61)	-	9063.4	1.99	60.66	2	3	31	10
P|016	HPS1 − 1Q	-	9172.08	1.09	−128.83	3	1	23	19
P|017	HPS1 +1Pyroglu	-	9283.32	1.27	−17.59	9	3	26	21
P|018	HPS1	9300.91	9301.17	1.07	0.25	1	3	79	70
P|019	HPS1 + 1(61)	-	9363.59	1.51	62.67	3	2	23	18
P|020	HPI2	10,654.46	10,653.97	1.01	−0.49	1	1	6	6
P|021	pre-P2	13,196.82	13,195.97	1.67	−0.85	1	4	58	51
P|022	pre-P2 + 1ph	-	13,275.94	1.80	79.12	3	4	27	22
P|023	pre-P2 + 2ph	-	13,355.92	1.54	159.1	1	5	28	29
P|024	pre-P2 + 3ph	-	13,435.64	1.59	238.82	3	3	14	17
P|025	pre-P2 + 4ph	-	13,515.5	1.19	318.68	2	2	8	6
P|026	Unknown	-	13,873.22	0.78	-		2		28
Unclustered	Unknown	-	-	-	-	57	89	166	94

**Table 2 proteomes-09-00021-t002:** Converging and diverging missassigments examples. Row (a) show an example of converging missassigment and row (b) its corresponding manual reassignation after cluster inspection: HPS2 +1(61) proteoform is more likely than HPS2. Rows (c) and (d) along with (e) and (f) provide two examples of diverging missassigments and their associated corrections. For both examples, HPS1 +1pyroGlu is more likely than HPS2 +2ph or HPS1 +1AcNterm, respectively. Check ProSight Lite fragmentation pattern, mass shift and remaining scoring columns for details (ph = phosphorylation, pyroGlu = pyroglutamic acid). Columns “m_e_” and “m_t_” give the experimental and theoretical masses, respectively. “Δm” gives the difference between m_e_ and m_t_. “P-Score” is the score as provided by ProSight Lite. “%Res. Cleav. is the percentage of inter-residues observed. “Match. Frag.” are the number of assigned matching fragments.

ProSight Lite Fragmentation Pattern	Form	Mod.	*m*_e_ (Da)	*m*_t_ (Da)	Δ*m*(Da)	P-Score	%Res. Cleav.	Match Frag.
(**a**) Originally assigned by PS 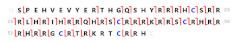 *BioRep A—TechRep 02—Scan 349*	HPS2	-	9063.83	9002.75	61.084	1.10 × 10^−91^	74	62
(**b**) Manual reassignation 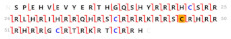 *BioRep A—TechRep R02—Scan 349*	HPS2	+1(61)	9063.83	9063.76	0.076	7.60·× 10^−155^	82	95
(**c**) Originally assigned by PS 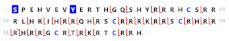 *BioRep A—TechRep R02—Scan 539*	HPS2	+2ph	9282.89	9162.68	120.213	7.90·× 10^−38^	52	35
(**d**) Manual reassignation 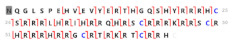 *BioRep A—TechRep R02—Scan 539*	HPS1	+1pyroGlu	9282.89	9283.89	−0.992	2.80·× 10^−103^	74	75
(**e**) Originally assigned by PS 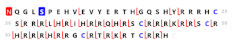 *BioRep B—TechRep R04—Scan 545*	HPS1	+1ph+1AcNterm	9283.87	9422.89	−139.022	9.70·× 10^−45^	59	40
(**f**) Manual reassignation 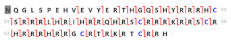 *BioRep B—TechRep R04—Scan 545*	HPS1	+1pyroGlu	9283.87	9283.89	−0.019	1.18·× 10^−118^	79	84

## Data Availability

The mass spectrometry proteomics data have been deposited to the ProteomeXchange Consortium via the PRIDE [39] partner repository with the dataset identifier PXD024405.

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
