# Peer review of "Protamine Characterization by Top-Down Proteomics: Boosting Proteoform Identification with DBSCAN"

_proteomes, 2021, doi:10.3390/proteomes9020021_

Round 1

Reviewer 1 Report

In this work Arauz-Garofalo et al, present a Top-Down analysis workflow combining two widely used software (Prosight PS -annotated database search- and TopPiC -un-annotated database search) in order to take the best of the two while overcoming their major weaknesses.

The text is particularly well written: the biological importance and challenges of studying protamines as well as the major bottlenecks of Top-Down proteomics are very well described in the introduction.

Marina Gay and coworkers identified 32 different protamine proteoforms, namely 13 proteoforms of P1 and 19 proteoforms of P2. They very precisely trouble-shoot both PS and TP results, and these observations (for example on converging and diverging missassigments) together with the pipeline they propose will be useful for the community. I recommend this manuscript for publication in Proteomes following minor corrections as follows:

Abstract :

“proteform” should read proteoform

Introduction:

“Including truncated” should read “including truncations”

Materials and methods:

2.1: Purification of human sperm protamine:

“fractions were dried at room temperature”: could you be more specific (use of speed-vacuum concentrator)?

2.4.1 TD Proteomics / LC-MS

Could you briefly precise the nature of the coupling with the Nanomate: the LC eluates were collected in the nanomate 96-well plate and then subsequently sprayed or did you use a particular coupling to spray directly from the nano-column ? If yes, at what flow-rate (250 nl/min?)

Please insert a scheme of the nanomate in Fig2. Precise the resolution of the MS2 scans in Fig.2

p5. “Such histones” should “read such as histones…”

Did you consider N-ter acetylation?

Why not including the truncated forms of P2 (HP2/HP3/HP4/HPI2/HPS1/HPS2) in your FASTA file since you expect them?

p6. “with all theoretically proteofoms” should read “with all theoretical proteoforms”

“The number of theoretical proteoforms in this first search was considerably reduced” could you be more specific (give an order of magnitude using equation 1).

During your first PS search without PTMs on the R residues, you did not find any acetylation nor methylation but could it be that you “missed” them precisely because you did not include the PTMs on the R residues ? If some proteoforms are modified both on STYM and R residues, would you find them?

It is not clear to me which nodes (Absolute Mass Search, biomarker?), which PTMs (on which residues) were used in the final three-tier search: could you please precise?

2.4.4: Proteoform Annotation

“whole-length” should read “full-length”

Table1. I am curious to know what is the standard deviation from the mean when calculating the average MW for each one of your clusters? Maybe that this value could indicate if the clustering went wrong and for some reasons included a wrong proteoform with a MW too different from the rest of the cluster (converging missasignement). Or maybe that this parameter is already taken into account when performing the clustering?

Cluster P-026: did you try to run a PS or TP search with an extended FASTA file containing major sperm–related proteins (without modifications on R residues to reduce the search space)?

I think you should comment that this +61 Da mass shift has already been seen in your TD previous studies of protamines or histones, as well as in other works (such as doi: 10.1021/acs.biochem.0c00293) in which the +61 Da adducts were clearly identified as copper binding via bicysteinate coordination.

It is not clear to me, whether you considered all Cys as alkylated when you calculated this +61 mass shift? Since you show later on, using ProsightLite, that this modification is indeed targeting Cys residues, the real mass shift (corresponding to this chemical modification) should be calculated by subtracting the experimental MW with a theoretical one including alkylations on all Cys except the one that is modified.

p.8: “web app” should read “web application”

General Remark: As stated, in p13, it appears that unclustered PrSMs correspond to low confidence matches. Fig3a/c or Fig.S2a clearly show that most of these unclustered PrSMs are coming from random MS2 events, since they do not show up as a straight line in the mass dimension. Is there a way to filter out (remove before the clustering) these single events/PrSM that do not cluster in terms of MW with any other ones? When reading you manuscript, I first understood that these 260 unclustered PrSM where corresponding to real proteoforms that could not be identified, whereas most of them are actually just artefacts or MS2 event that were triggered randomly.

The Fig3c (benefit of DBScan over PS and TP) could be more striking if you used colors that are more different for each cluster.

3.4 Modification site assignment:

Could you please comment on what is known from the literature concerning the localization of these p-sites on P1 and P2?

Overall, the advantage of SBSCAN over PS and TP seems quite evident. I am looking forward to see this pipeline applied the TD characterization of protamines from abnormal spermatozoids and to see if a ratio more accurate than just the P1/P2 ratio could be used. Or maybe specific proteoforms could be used to diagnose certain specific mal infertility cases.

The PXD number of the repository is stated in the text, but the authors should provide the login and password to the reviewer for inspection of the deposited dataset.

Reviewer 2 Report

The authors performed a top-down proteomic investigation of two normozoospermic sperm samples to identify protamine proteoforms at different levels. They used two software packages, the ProSight PD (PS) and the TopPIC (TP) suite, with a clustering algorithm to decipher protamine proteoforms. The article is interesting for specialized readers in the field. However, the analysis of only two normozoospermic samples represents the major limitations of the manuscript, which makes the manuscript a method description and not an original article.

Major 

  1. Authors should perform a top-down proteomics analysis on sperm samples with morphological head abnormalities, which could indicate abnormal DNA compacting. Only in this way, they could apply this identification method for personalized therapy. 
  2. How it is possible to quantify the expression levels of identified proteoforms?

  This in-depth analysis of the protamine proteoform landscape of an individual boosts the personalized diagnosis of
male infertility

Reviewer 3 Report

In this paper, Authors aimed to assess the usefulness of the DBSCAN algorithm to solve some of the misassignments found when analyzing the top-down database searches. I found this paper interesting and scientifically sound and has merit of publication.

Annotated points:

-how findings can be translated to routine? Could be advantageous perform the analysis in different biological samples instead technical replicates, which number is quite low.

-some illustrative spectra showing PTM localization probability.

- A resume integrative picture could be done to provide an illustration of data and transpose it.

Round 2

Reviewer 2 Report

Authors justify their choose to analyze only two normospermic samples. Although it is interesting for readers, it is the reviewer's opinion that the paper lacks of a comparison. In fact, since the low number of analyzed samples , a major criticism could be to evaluate the variability between individuals. I suggest to perform new experiments. If the editor approve the paper publication I suggest to discuss the paper limitations in the discussion section

Reviewer 3 Report

 Authors have addressed my comments.
